# The Benefits, Risks and Regulation of Using ChatGPT in Chinese Academia: A Content Analysis

Jason Hung [1],* and Jackson Chen [2]

[1] Department of Sociology, The University of Cambridge, Cambridge CB2 1SB, UK
[2] Department of Sociology, The London School of Economics, London WC2A 2AE, UK;
jacksonchen30@yahoo.com
* Correspondence: ysh26@cam.ac.uk

**Abstract: Research Aims.** This research project investigates what are the major benefits and risks of Chinese students using ChatGPT for academic activities. Also, the project assesses how, if applicable, should ChatGPT be regulated in Chinese academic settings in order to maintain academic integrity and ethical standards. **Methodology.** The collection of primary data from relevant newspaper articles serves as the foundation of this research project. Here, the content analysis is used for primary data collection. A combination of keywords ["ChatGPT" AND ("China" OR "Chinese") AND ("students" OR "student")] were typed on the Google news search engine on 12 April 2023. A sum of 40 newspaper articles were deemed eligible for data analysis. Both qualitative and quantitative data were extracted and analyzed. **Findings.** The opinion of using ChatGPT to fulfil academic responsibilities has been polarized in China. The conservative camps worry that students are using ChatGPT to commit academic cheating. However, some Chinese educators believe AI-powered technologies should be incorporated into academic learning as AI-enabled writing tools can help improve the quality of academic outputs. A major concern that Chinese educators hold, to date, is plagiarism violations by students as an act of academic cheating. Most newspaper articles presented the use of ChatGPT in Chinese academic settings neutrally. Newspaper articles published in March 2023 contained more positive and negative word(s) about the use of ChatGPT in academic learning. **Conclusions.** Given the benefits ChatGPT can provide and the near-infeasibility of massively ban the use of AI-powered software, more regulations should be set up in Chinese academia. Teachers have to guide students on how to fact-check the details provided by AI and add citations and references accordingly in their coursework. Moreover, teachers should guide students on how to ask AI-powered software questions systematically and creatively, in order to maximize the intellectual outputs generated from ChatGPT.

**Keywords:** ChatGPT; artificial intelligence; academic learning; education; integrity; China

## 1. Introduction

ChatGPT is a human–artificial intelligence (AI) interaction application that was released on 30 April 2022 (Heaven 2023). Since then, ChatGPT has been popularized worldwide rapidly (George and George 2023). ChatGPT, in part, is capable of answering a wide variety of human-asked questions in ample domains of knowledge. ChatGPT also contains a function that automatically writes an essay in fewer than minutes (Biswas 2023). ChatGPT is, therefore, seen as a revolutionized instrument that helps students complete their coursework and assessments (Rudolph et al. 2023). It is controversial whether students should be enabled to use ChatGPT to complete any school assignments, especially when such a form of academic assistance may be deemed as an act of cheating that violates academic integrity.

With the use of ChatGPT, students can be provided with personalized, reliable and specialized guidance in completing schoolwork on behalf of the academic learners themselves. Ranging from a subject-specific query to a general academic issue, ChatGPT can

deliver a tailor-made response/solution in fewer than minutes, reducing the amount of human effort to complete school assignments to a significant extent. ChatGPT, in addition, is always accessible. Regardless of the time zones and locations users follow or reside in, respectively, they do not face any constraints that bar them from using ChatGPT to help complete coursework or assessments (Baidoo and Ansah 2023). As many ChatGPT applications are either freely accessible or require a very small payment to use, instrumentalizing such technology to facilitate academic fulfilment is a cost-effective approach (Mhlanga 2023). Even if students are not financially independent or advantaged, they should be able to afford the use of ChatGPT to assist in their completion of academic tasks.

ChatGPT also contains ample languages as media of instructions. Even if students need to complete their academic tasks in languages other than English, the application should be helpful to provide instant academic advice and assistance. Not only is ChatGPT a linguistically compatible application, it is also culturally inclusive that helps answer queries based on cultural contexts (Kasneci et al. 2023). As a result, the convenience, accessibility. and inclusivity of ChatGPT have propelled its popularity in academic learning. It is noteworthy that ChatGPT, in addition to answering academic questions, can edit and proofread academic essays (Lecler et al. 2023). Users are able to benefit from ChatGPT from the early stages of research and planning to the final stage of preparing their work for submission. Given all these benefits, not only is the rapid popularization of ChatGPT sparking international attention but concerns about academic and research integrity have been raised owing to how the revolutionized application interrupts students' academic dedication (Shiri 2023).

This research project, therefore, addresses the academic concerns and risks of using ChatGPT to help complete coursework and assignments. As of 13 January 2023, ChatGPT has by far been most popular in China, followed by Nepal, Norway and Singapore, per the number of "ChatGPT" searches on the Google search engine (Baltrusaltis 2023). Given ChatGPT's disproportionately high popularity in the China market, this research paper focuses on evaluating the academic concerns and risks of the use of the application in Chinese contexts. The research project involves the collection, summarization and management and analysis of first-hand data concerning the use of ChatGPT in Chinese academic settings. Findings will be presented in order to unveil how the emergence and popularity of ChatGPT have been viewed in China's academia. To date, there is an absence of similar research projects in Chinese academic contexts. Therefore, the findings of this research project contain scholarly originality and significance, which should be insightful for academics within and even beyond the China market to consider how ChatGPT should be regulated in future.

## 2. Literature Review

### 2.1. Risks of Using ChatGPT

As an AI-powered interactive platform, ChatGPT offers ample benefits that are conducive to students' learning, development and self-growth, so far as such technology is appropriately used (Yan 2023). However, when ChatGPT is not used aptly, a range of risks may be posed that harm the interests of students, teachers and schools.

A primary concern is that students overly relying on ChatGPT to complete essays and coursework may be at risk of violating the plagiarism rules (Anders 2023). ChatGPT generates responses to the queries of students based on existing data, there is a high possibility that students using the application to complete academic work would inadvertently include content that is copied from existing, published sources (Choi et al. 2023). Those violating plagiarism are accused of significant academic misconduct and may potentially be subject to school punishment. Students are supposed to, at most, use ChatGPT as an assistance tool to help guide their thinking and research process in order to help facilitate their own development of essays or other academic work. However, given the laziness and irresponsibility of a portion of students, these ChatGPT users may directly copy the AI-generated responses to answer academic assessment questions (Khalil and Er 2023).

Not only do these students lose the opportunity to develop their critical thinking and independent learning skills, but they are also performing academic misconduct that should be punitively prohibited.

These students, over time, may be increasingly dependent on ChatGPT for the completion of their academic assignments. These students, in the long run, lose the opportunity to accumulate their pool of knowledge and problem-solving ability. Being overly dependent on ChatGPT to complete assessments, in addition, hinders students from building their creativity and long-term cognitive development, harming their overall academic, social and career performance if ChatGPT is not applicable in any setting (Rudolph et al. 2023).

Those abusing ChatGPT as a shortcut to completing school tasks may miss out on valuable opportunities to develop their collaboration and communication skills too. With the availability of ChatGPT, students may be less willing to seek help from their teachers, peers or parents, as they believe the AI-powered programme is able to come up with a (non-)academic solution instantly. The loss of opportunities to sharpen their collaboration and communication skills limits their ability to work effectively in groups—a needed criterion for success in ample professional fields (Arif et al. 2023).

Using ChatGPT aptly can help academic learners complete their tasks more cost-effectively and in a timely manner. However, when academic learners unduly depend on the AI-powered platform to complete academic assignments, their levels of independence are likely to diminish. Therefore, the use of ChatGPT, despite the notable benefits, has been sparking international concern.

### 2.2. Should ChatGPT Be Regulated?

These concerns lead to the consideration of whether the use of ChatGPT should be tightly regulated. In today's climate, online education has been expanding massively (Firat 2023). The use of technological platforms to facilitate academic learning and development has been growingly common. However, why the use of ChatGPT has been drawing a raft of controversies is because, unlike alternative technological tools, users may exploit such an AI-powered tool to challenge academic integrity. The purpose of academic learning is not to achieve impressively high scores, but to gain critical thinking and problem-solving skills that are conducive to students' lifelong development. If academic learners solely or primarily rely on ChatGPT to complete coursework and assignments, they are wasting the opportunities to develop these skills in favor of their development and growth (Iskender 2023).

Inappropriately using ChatGPT also poses ethical considerations (Sallam 2023). As said, some students may lazily cheat on coursework or assignments by submitting plagiarized work. Such an ethical issue violates the purpose of learning and damages the academic conduct that all students should comply with in educational settings. At higher-level education, students may logically be given more rights and higher flexibility to use ChatGPT when completing their academic work. However, at lower levels, students may not necessarily have the maturity and awareness of how to aptly use AI-powered technology while, simultaneously, building their own knowledge pool and independent thinking, critical thinking and problem-solving skills.

While ChatGPT is a helpful tool to enhance students' academic learning progress and efficiency, the technological tool should not be used as a subtitle for academic diligence, problem-solving and critical thinking (Dis et al. 2023). Appropriate policies should be set up in academia accordingly to help educators ensure that academic and ethical integrity is maintained by students.

### 3. Research Aims and Questions

This research project investigates what are the major benefits and risks of Chinese students using ChatGPT for academic activities. Also, the project assesses how, if applicable, should ChatGPT be regulated in Chinese academic settings in order to maintain academic integrity and ethical standards. Research questions are written as follows:

1. Does the use of ChatGPT pose any benefits for Chinese students' academic learning?
2. Does the use of ChatGPT pose any risks to Chinese students' academic learning?
3. If applicable, how should educators regulate the use of ChatGPT in Chinese academic settings?

## 4. Methodology

### 4.1. Data Collection

The collection of primary data from relevant newspaper articles serves as the foundation of this research project. Here, the content analysis is used for primary data collection. The combination of the keywords ["ChatGPT" AND ("China" OR "Chinese") AND ("students" OR "student")] was typed on the Google news search engine on 12 April 2023. Here, a total of 55 newspaper articles were displayed. However, 15 of the newspaper articles did not contain the word "China" or "Chinese" in the title, subtitle(s) or main text, but the word "China" or "Chinese" appeared in the commercial advertisement(s) attached to the news pieces. These 15 articles were excluded. Therefore, only 40 newspaper articles were deemed eligible for data analysis. A sum of 13 newspaper articles were published by South China Morning Post, and four were from China Daily, three from Reuters, two from Gizchina, one from Times Higher Education, one from Study International, one from Business Telegraph, one from Verve Times, one from Sky News, one from France24, one from Interesting Engineering, one from Global Times, one from Bloomberg, one from CGTN, one from the Brown Daily Herald, one from Tech Business News Australia, one from Nation World News, one from Money Control, one from Tech Juice, one from Tech Times, one from Business Today, one from India Times and one from India Today. The articles were published between 14 December 2022 and 11 April 2023.

### 4.2. Data Management and Analysis

Data extracted from the eligible newspaper articles were inputted into a Microsoft Excel file. Here, information including the corresponding newspaper outlet(s), and publication date(s) of the newspaper article(s) was entered into the Excel file for data management. The 40 eligible newspaper articles were read thoroughly. Here, qualitative and quantitative data were extracted, forming the basis of the findings of this research project.

For qualitative data, sentences addressing the benefits/risks of Chinese students using ChatGPT to help complete coursework or assignments were extracted directly from eligible newspaper articles. Also, sentences delineating how major stakeholders (such as the education bureau(s), school(s)/university(ies) or student(s)) should respond to the use of ChatGPT to help regulate such a technological tool/platform were extracted directly from the newspaper articles.

For quantitative data, each of these 40 newspaper articles was read and analyzed in-depth. Each newspaper article was categorized as presenting a (1) positive, (2) neutral or (3) negative theme of the use of ChatGPT in Chinese academic settings. For example, the use of ChatGPT being conducive to improving the efficiency in academic learning and the use of ChatGPT being beneficial in developing students' time management capacity are examples of positive themes. The use of ChatGPT being detrimental to students' development of critical and independent thinking skills and putting students at risk of violating academic plagiarism are examples of negative themes. Articles that solely outline the function of ChatGPT in academic learning settings without indicating whether such an AI-powered instrument is beneficial or detrimental to students' development is an instance of presenting a neutral theme. Moreover, the number of positive/negative words/phrases referring to the use of ChatGPT in each newspaper article were recorded. For example, improv[ing] learning efficiency and time management are positive phrases; undermin[ing] critical thinking skills and/or independent thinking skills, impos[ing] risk of cheating and violat[ing] academic plagiarisms are negative phrases. J.H. invited J.C. for both to independently screen all eligible newspaper articles to quantify the number of positive and negative word(s)/phrase(s) included and the number of positive, neutral and negative

theme(s) presented. Screening results developed independently by J.H. and J.C. were highly similar. Results that differed were discussed between J.H. and J.C. in order to make a final decision on how such screening outcomes would be treated. Upon finalizing the quantitative interpretation of screening outcomes, these quantitative data were stored in the Excel file for better data management. All quantitative data were transferred to the software package STAT 17.2 for the production of descriptive statistics. For example, an analysis of variance (ANOVA) was performed. The ANOVA outputs are presented in tables in the following sections.

## 5. Findings

### 5.1. The Benefits of Using ChatGPT in Academic Learning

Table 1 summarizes the basic information of all eligible newspaper articles that were included, studied and analyzed. Ever since the launch of Open AI's ChatGPT, Chinese students have been using this technological tool to complete their academic assignments (such as writing book reports and practicing language skills) (Tamim 2023). The opinion of using ChatGPT to fulfil academic responsibilities has been polarized in China. The conservative camps worry that students are using ChatGPT to commit academic cheating. However, some Chinese educators believe AI-powered technologies should be incorporated into academic learning, as AI-enabled writing tools can help improve the quality of academic outputs (Keyue 2023). Also, ChaGPT helps users gather reference materials for their research papers or essays, significantly increasing the efficiency of students' work and allowing them to devote more energy and time to creative tasks (Keyue 2023).

**Table 1.** Summary of eligible newspaper articles.

| Author (Year) | Newspaper Outlet | Title |
|---|---|---|
| Shen (2023) | *South China Morning Post* | China's Internet Watchdog Proposes Rules, Security Assessment for AI Tools Similar to ChatGPT |
| Tamim (2023) | *Interesting Engineering* | 'Great Firewall': Chinese Students Use ChatGPT to Cheat on School Work |
| *AFP News* (2023) | *France24* | China's Students Leap 'Great Firewall' to Get Homework Help from ChatGPT |
| Nan and Chu (2023) | *China Daily* | ChatGPT Can Help Enrich Higher Education |
| Booth (2023) | *Verve Times* | China's ChatGPT Black Market Is Thriving |
| Abdullah (2023) | *Gizchina* | Students No Longer Copy: They Use ChatGPT |
| J. Liu (2023) | *Times Higher Education* | How Will Chinese Universities Respond to the Rise of ChatGPT? |
| Baltrusaltis (2023) | *India Times* | ChatGPT Demand on Google Hits a Record High as China Dominates Interest |
| *Reuters* (2023b) | *Reuters* | China Says It Sees the Potential of ChatGPT-Like Technology |
| Hew (2023) | *Study International* | Don't Use ChatGPT. You Could Lose Your Study Visa |
| *Money Control News* (2023a) | *Business Telegraph* | AI News Roundup: Chinese Students Lean on ChatGPT, Google Building USM to Compete with Microsoft and More |
| Chandra (2023) | *South China Morning Post* | How Academia Can Embrace ChatGPT and Reignite a Love for Learning |
| Acres (2023) | *Sky News* | Alibaba Enters AI Arms Race with ChatGPT-Like Model Named Tongyi Qianwen |
| *South China Morning Post* (2023a) | *South China Morning Post* | A Few Quick Fixes to Deter ChatGPT-Enabled Cheating |
| E. Udin (Chakravarti 2023) | *India Today* | Using ChatGPT Is Now a Punishable Offence in this University, Know Why |

**Table 1.** *Cont.*

| Author (Year) | Newspaper Outlet | Title |
| --- | --- | --- |
| *South China Morning Post* (2023e) | *South China Morning Post* | Chinese University of Hong Kong Students to be Expelled If Caught Using ChatGPT or AI Tools, New Guide Reveals |
| Nation World News (2023) | *Nation World News* | ChatGPT is Banned in China, But Students Love It |
| South China Morning Post (2023f) | *South China Morning Post* | How Universities Can Start to Grapple with ChatGPT's Capabilities |
| *South China Morning Post* (2023b) | *South China Morning Post* | AI Should Be Embraced for Learning While Still Guarding against Abuse |
| He (2023) | *China Daily* | How Can Academia Combat ChatGPT? |
| O. Liu (2023) | *South China Morning Post* | Hong Kong's Former Finance Chief John Tsang Slams Universities' ChatGPT Ban, Saying If They Can Use Calculators, They Can Use AI |
| Shuo (2023) | *China Daily* | ChatGPT's Arrival Provokes Thought on Future of Education |
| *Money Control News* (2023b) | *Money Control* | Weekend AI News Roundup: AI in BPOs, Chinese Students Accessing ChatGPT, Inker Robotics Secures Million Dollar Funding, More |
| Khursheed (2023) | *Tech Juice* | Chinese Students Seek Homework Assistance from ChatGPT |
| Keyue (2023) | *Global Times* | Papers Concealing Usage of ChatGPT to Be Rejected or Withdrawn |
| Cassidy (2023) | *The Guardian* | College Student Claims App Can Detect Essays Written by Chatbot ChatGPT |
| Mudbhary (2023) | *The Brown Daily Herald* | International Students Share Opinions on ChatGPT in Home Countries |
| Richard (2023) | *Tech Times* | ChatGPT via WeChat Used by Chinese Students in School, Learn English—Should it be Normalized? |
| Stone (2022) | *Bloomberg* | Anti-Cheating Education Software Braces for AI Chatbots |
| *CGTN* (2023) | *CGTN* | University of Hong Kong Issues Interim Ban on ChatGPT, AI-Based Tools |
| *Tech Business News Australia* (2023) | *Tech Business News Australia* | China Leads in ChatGPT Demand Despite Reports Tech Giants Expressing Opposition |
| *South China Morning Post* (2023d) | *South China Morning Post* | ChatGPT's Rise Highlights Need for More Support of Open Source Community |
| Yiu (2023) | *South China Morning Post* | 2 Universities in Hong Kong Embrace Use of ChatGPT, Other AI Tools to Boost Quality of Teaching, Learning |
| Mok (2023) | *South China Morning Post* | Hong Kong's Baptist University Bans Students from Using ChatGPT for Class Work |
| Fung (2023) | *South China Morning Post* | Is ChatGPT the Future of Learning? Educators, Students Discuss How Schools Can Adapt to Potential of AI-Driven World |
| D'Cruze (2023) | *Business Today* | 'Chinese ChatGPT' to Launch Soon to Take on OpenAI and Google |
| Udin (2023) | *Gizchina* | ChatGPT Effect: How to Prevent Students from Cheating/Copying |
| *South China Morning Post* (2023c) | *South China Morning Post* | ChatGPT Makes Me Appreciate Authors and My Friends |
| *Reuters* (2023a) | *Reuters* | ChatGPT Mania Pumps up Chinese AI Technology Stocks |
| *Reuters* (2023c) | *Reuters* | Chinese State Media, AI Companies Warn of Risks in ChatGPT Stock Frenzy |

Time spent by Chinese students on homework has been reduced notably due to the use of ChatGPT. They are, simultaneously, keen on using the application to learn English by interacting with the AI assistance. They can even use ChatGPT to solve complicated science and mathematics problems, in addition to generating computer code (*AFP News* 2023). These benefits help enhance learning efficiency if Chinese students use ChatGPT aptly. Teachers, moreover, have been using ChatGPT to generate customized lesson plans

in a few seconds. Such a benefit helps shorten the administrative workload that teachers are conventionally tied up with, allowing them to enjoy better work efficiency (Tamim 2023). With the use of ChatGPT, teachers can efficiently create lesson plans and study materials for their students in a desirable and cost-effective manner. Repetitive work that teachers used to find time-consuming to complete is now being performed by ChatGPT (Richard 2023).

The University of Hong Kong and Hong Kong Baptist Universities are among the first Chinese universities that have banned students' use of ChatGPT or its AI-powered equivalents for the completion of coursework and assignments (J. Liu 2023). However, there are dissidents arguing that the ban on the use of ChatGPT is unnecessary as it "disallow[s] the use of calculators when doing mathematics" (O. Liu 2023). Dissidents believe ChatGPT should be allowed to help students write the first draft of their coursework and assignments. Students can, then, tweak the sentence structures and words of the AI-written draft and add in-depth analysis to enrich the content (O. Liu 2023).

The Chinese University of Hong Kong seconds the decision of banning the use of ChatGPT by emphasizing that misuse of ChatGPT could result in school expulsion. The Hong Kong University of Science and Technology, however, embraces the use of ChatGPT, allowing both staff and students to use the application to help them complete school tasks (O. Liu 2023). "Generative AI tools are not going to go away. They will adapt and become more advanced . . . it is our responsibility as educators to prepare students for an AI-driven world and future work," said Sean McMinn, the Director of the Centre for Education Innovation at the Hong Kong University of Science and Technology (Fung 2023).

*5.2. The Risks of Using ChatGPT in Academic Learning*

A major concern Chinese educators hold, to date, is plagiarism violations by students as an act of academic cheating (Abdullah 2023). ChatGPT's answers can easily be identified by teachers and Turnitin—arguably the most popular academic plagiarism-checking application—software. Moreover, ChatGPT's answers make ample factual errors. For example, ChatGPT modified the publication year of a source from the 1990s to 2011 in order to meet the requirement to cite recent sources in an essay (*South China Morning Post* 2023f). Despite the answers being viewed convincingly correct, many factual errors actually occur when students use ChatGPT to produce any answers (Stone 2022).

Chinese educators worry that ChatGPT could change the fundamental training structure of higher education. In China, students, conventionally, take 16 years of full-time learning to earn a bachelor's degree and over 20 years of full-time learning to obtain a PhD degree. Some two decades of learning enable students to accumulate a storehouse of knowledge. However, with the emergence and popularization of ChatGPT, the conventional academic learning system is at risk of being replaced by AI-powered software. Chinese educators and innovators remain uncertain, to some degree, about how Chinese academia should and could transition from offering decades-long, human-centric learning opportunities to incorporating AI-powered software to assist students' development of academic and personal growth (Nan and Chu 2023). With the use of ChatGPT, AI, despite its convenience and cost-effectiveness, nurtures a very limited way of perceiving knowledge, society and culture, which could be detrimental to human intellectual development in the long term. Under the AI-powered epoch, without ensuring that students can learn the importance of criticizing texts created by AI by developing independent thinking skills, educators face many ethical and practical challenges that have to be solved prior to further promoting students' use of ChatGPT in academic learning (*South China Morning Post* 2023f). Today, Chinese students using ChatGPT to write coursework and assignments, once caught, could be penalized by grade reduction, course failure and school suspension or dismal (Hew 2023; *South China Morning Post* 2023e). However, without a mature, well-designed regulatory system and given the convenience and popularity of using ChatGPT to complete academic work, very unlikely adding punitive terms against the use of AI-powered software can deter the majority of students from capitalizing on AI tools to realize academic fulfilment. Educators have to, in response, develop clear guidelines on how ChatGPT can be autho-

rized for use by students as soon as possible, in order to facilitate academic learning while maintaining academic and ethical integrity (*South China Morning Post* 2023a). At the University of Hong Kong, for example, while students are in general banned from using ChatGPT to complete academic tasks, they may seek written consent from their course instructors, where requests are evaluated on a case-by-case basis, to detail why and how they decide to use ChatGPT to assist their academic fulfilment (*CGTN* 2023; Chakravarti 2023).

### 5.3. How Should the Use of ChatGPT Be Regulated in Academic Settings

Possible regulations to control the use of ChatGPT and maintain academic and ethical integrity could be setting the requirement of oral defenses of students' essays. A panel of examiners could be formed to randomly draw students from each class and invite them to attend oral defenses. Oral defenses are a conventional pedagogical method that helps determine whether students wrote the essays or assignments by themselves in order to deter the act of academic cheating (*South China Morning Post* 2023a).

Teaching staff can also revisit the old-school assessment method of holding an examination that requires the development of written essays. Students could be asked to produce short essays immediately on a pre-scheduled date in an examination room without permission to bring any digital devices into the assessment settings. Such a conventional method helps teaching staff evaluate how much students have learned and retained over the semester or term (*South China Morning Post* 2023a).

In the long term, teaching staff can grade students by their learning progress rather than by their end-of-semester academic performance. For example, more assessment-based tutorial discussions should be held during the semester. More tutorial sessions, in lieu of lectures, should be held to allow students to express their views and ask questions. Teaching staff should, then, assess students' academic performance throughout the semester (*South China Morning Post* 2023a).

The future is about integrating technology with humans. The emergence and popularization of ChatGPT propel educators to consider how pedagogy and learning should be evaluated and updated (Shuo 2023). Educators have to identify what are the purposes of education and academic learning, in order to update the learning regulations that help ensure these purposes per se can be preserved (Shuo 2023).

### 5.4. The Trajectory of Media Narrative on ChatGPT Use in Chinese Academic Settings

This research project, in addition to collecting qualitative, textual data, quantitatively analyzed (1) the newspaper themes on the use of ChatGPT in academia, (2) the number of positive words/phrase(s) contained in each newspaper article about the use of ChatGPT in academia, and (3) the number of negative words/phrase(s) included in each newspaper articles about the use of ChatGPT in academia.

Findings demonstrate that 75 per cent of newspaper articles presented the use of ChatGPT in academia neutrally. However, there were 15 per cent and 10 per cent of newspaper articles delineating ChatGPT use in academia as negative and positive, respectively (Table 2).

**Table 2.** Newspaper theme about ChatGPT by month, December 2022 to April 2023.

| Theme/Date | December 2022 | January 2023 | February 2023 | March 2023 | April 2023 | Total |
|---|---|---|---|---|---|---|
| Positive | 0 (0.00%) | 0 (0.00%) | 1 (10.00%) | 2 (9.52%) | 1 (33.33%) | 4 (10.00%) |
| Neutral | 1 (100.00%) | 5 (100.00%) | 6 (60.00%) | 16 (76.19%) | 2 (66.67%) | 30 (75.00%) |
| Negative | 0 (0.00%) | 0 (0.00%) | 3 (30.00%) | 3 (14.29%) | 0 (0.00%) | 6 (15.00%) |
| Total | 1 (100.00%) | 5 (100.00%) | 10 (100.00%) | 21 (100.00%) | 3 (100.00%) | 40 (100.00%) |

A sum of 18 newspaper articles contained positive word(s)/phrase(s) about the use of ChatGPT in academia. Two-thirds of them mentioned positive word(s)/phrase(s) about ChatGPT use once or twice per article (Table 3). These word(s)/phrase(s) include "im-

prov[ing] learning efficiency", "allow[ing] . . . understanding of different subjects", "[the outputs of ChatGPT being] academically sound" and "[the outputs of ChatGPT being mostly] accurate". There were two outliers that each mentioned positive word(s)/phrase(s) about the use of ChatGPT academically five and ten times. Most articles presented the advantages of using ChatGPT in academia in March 2023 (Table 3).

**Table 3.** Number of Positive Words about ChatGPT by Month, December 2022 to April 2023.

| Positive Words/Date | December 2022 | January 2023 | February 2023 | March2023 | April 2023 | Total |
|---|---|---|---|---|---|---|
| 0 | 1 (100.00%) | 2 (40.00%) | 7 (70.00%) | 10 (47.62%) | 2 (66.67%) | 22 (55.00%) |
| 1 | 0 (0.00%) | 1 (20.00%) | 1 (10.00%) | 4 (19.05%) | 0 (0.00%) | 6 (15.00%) |
| 2 | 0 (0.00%) | 1 (20.00%) | 0 (0.00%) | 4 (19.05%) | 1 (33.33%) | 6 (15.00%) |
| 3 | 0 (0.00%) | 1 (20.00%) | 1 (10.00%) | 2 (9.52%) | 0 (0.00%) | 4 (10.00%) |
| 5 | 0 (0.00%) | 0 (0.00%) | 1 (10.00%) | 0 (0.00%) | 0 (0.00%) | 1 (2.50%) |
| 10 | 0 (0.00%) | 0 (0.00%) | 0 (0.00%) | 1 (4.76%) | 0 (0.00%) | 1 (2.50%) |
| Total | 1 (100.00%) | 5 (100.00%) | 10 (100.00%) | 21 (100.00%) | 3 (100.00%) | 40 (100.00%) |

A total of 17 newspaper articles encompassed negative words(s)/phrase(s) about the use of ChatGPT in academia in the main text. A sum of 76.5 per cent of them mentioned negative word(s)/phrase(s) about the ChatGPT use one to three time(s) per article (Table 4). These word(s)/phrase(s) include "violat[ing] academic plagarism", "cheating", "undermin[ing] critical and independent thinking" and "lazi[ness]". There were three outliers that mentioned negative words/phrase(s) about the use of ChatGPT academically five or six times each. Most articles delineated the disadvantages of using ChatGPT in academia in March 2023 (Table 4).

**Table 4.** Number of negative words about ChatGPT by month, December 2022 to April 2023.

| Positive Words/Date | December 2022 | January 2023 | February 2023 | March 2023 | April 2023 | Total |
|---|---|---|---|---|---|---|
| 0 | 1 (100.00%) | 3 (60.00%) | 6 (60.00%) | 11 (52.38%) | 2 (66.67%) | 23 (57.50%) |
| 1 | 0 (0.00%) | 1 (20.00%) | 1 (10.00%) | 2 (9.52%) | 0 (0.00%) | 4 (10.00%) |
| 2 | 0 (0.00%) | 0 (0.00%) | 0 (0.00%) | 5 (23.81%) | 0 (33.33%) | 5 (12.50%) |
| 3 | 0 (0.00%) | 1 (20.00%) | 1 (10.00%) | 1 (4.76%) | 1 (33.33%) | 4 (10.00%) |
| 4 | 0 (0.00%) | 0 (0.00%) | 1 (10.00%) | 0 (0.00%) | 0 (0.00%) | 1 (2.50%) |
| 5 | 0 (0.00%) | 0 (0.00%) | 1 (10.00%) | 0 (0.00%) | 0 (0.00%) | 1 (2.50%) |
| 6 | 0 (0.00%) | 0 (0.00%) | 0 (0.00%) | 2 (9.52%) | 0 (0.00%) | 2 (5.00%) |
| Total | 1 (100.00%) | 5 (100.00%) | 10 (100.00%) | 21 (100.00%) | 3 (100.00%) | 40 (100.00%) |

Figure 1 shows that, in most circumstances, the media theme on the use of ChatGPT academically is neutral (Figure 1). However, for newspaper articles published in February and March 2023, media theme on the use of ChatGPT in academia tended to be observably negative. For newspaper articles published in April 2023, alternatively, the average of the media theme was inclined to be moderately positive (Figure 1).

For newspaper articles published in March 2023, each news piece on average contained almost 1.4 positive words referring to the use of ChatGPT in academic settings (Figure 2). In addition, each news piece on average encompassed some 1.3 negative words concerning ChatGPT use in academia in both February 2023 and March 2023 (Figure 3). For articles published in January 2023, each news piece on average had more positive than negative words about ChatGPT use in academic settings (Figures 2 and 3). However, for articles published in April 2023, each news piece on average carried more negative than positive words about the use of Chat GPT academically (Figures 2 and 3).

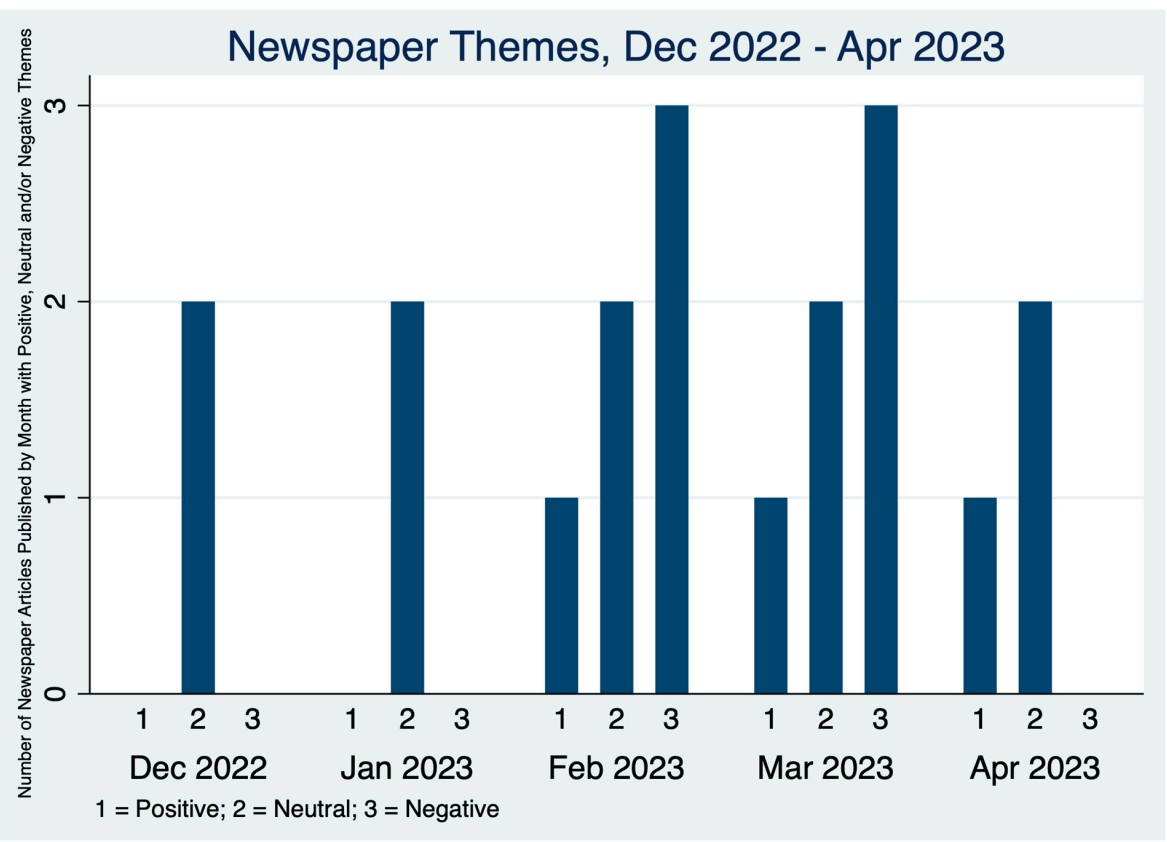

**Figure 1.** Newspaper theme about ChatGPT by month, December 2022 to April 2023.

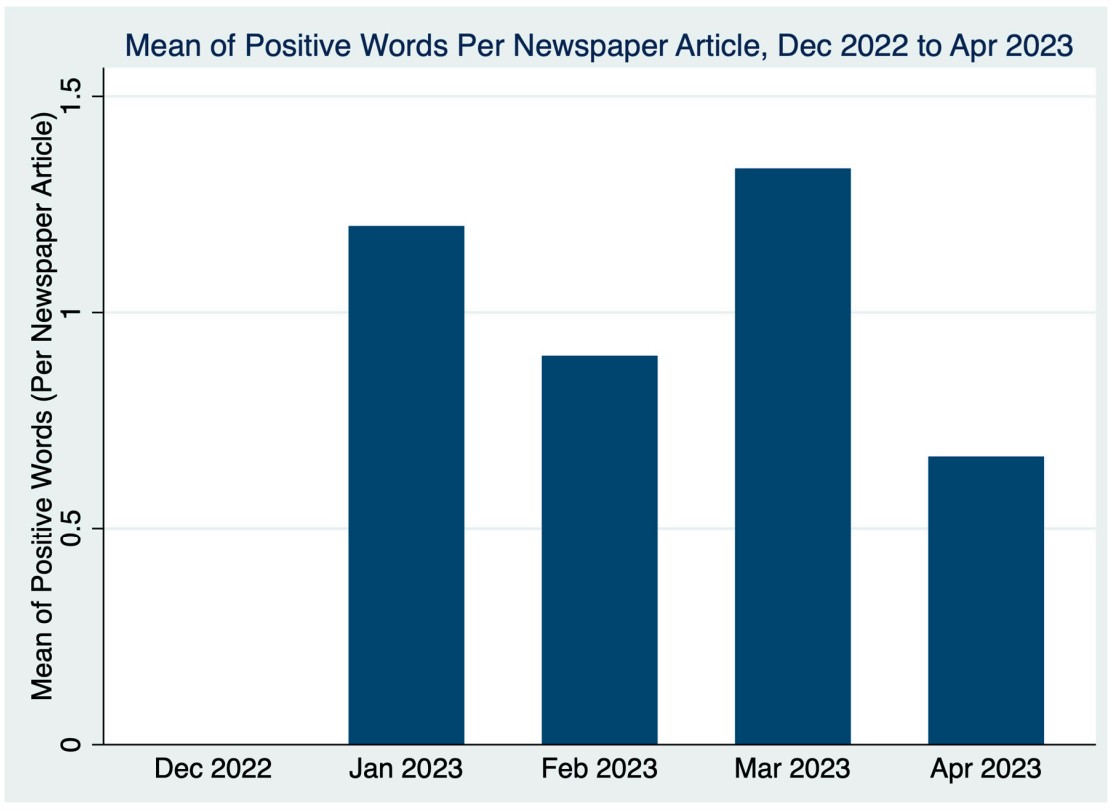

**Figure 2.** Number of positive words per article about ChatGPT by month, December 2022 to April 2023.

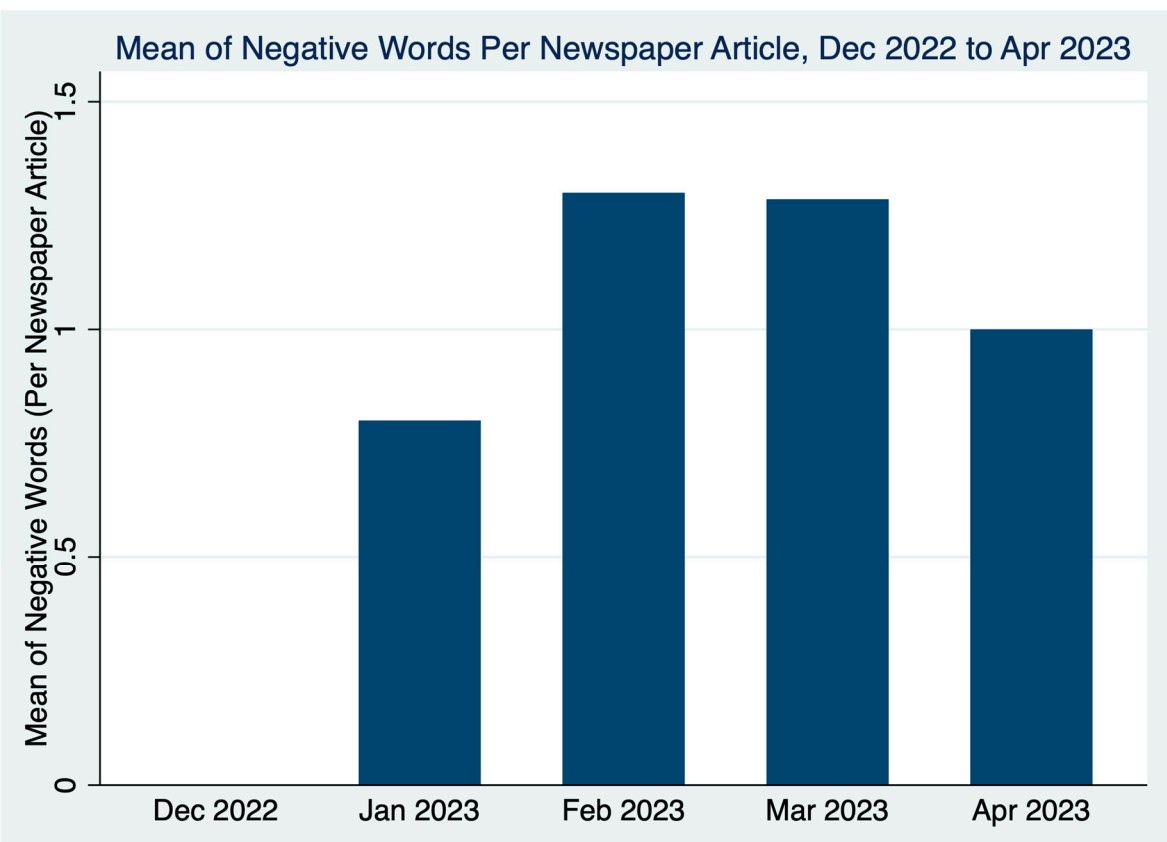

**Figure 3.** Number of negative words per article about ChatGPT by month, December 2022 to April 2023.

## 6. Discussion

Opportunities always come with risks. ChatGPT is a revolutionized software that is easily accessible, user-friendly and compatible with academic learning. However, any misuse of ChatGPT may violate academic integrity and ethical concerns. Also, such misuse discourages students, who are still cognitively developing, from building their critical thinking, independent thinking and problem-solving skills. ChatGPT is a useful tool that helps minimize both students' and teachers' time and effort spent on academic learning or instructions, so far as the AI-powered software is aptly used. As said, higher-grade students should be given more flexibility and higher capacity to use of ChatGPT in academic learning, given the fact that they are more independent and mature as learners. However, for lower-grade students, their use of ChatGPT should be tightly regulated in order to avoid any over-reliance on AI-powered software to help them complete academic tasks.

ChatGPT, as mentioned, is also a beneficial tool that helps students develop linguistic and cultural competence. Such features should be highlighted when promoting academic learning. For example, for non-native English speakers, teachers should guide students on capitalizing on the use of ChatGPT to engage in AI-human interactive or intellectual exchange opportunities for English learning. Also, for students who would like to enrich their own cultural understanding, they can capitalize on the use of ChatGPT by asking AI about their culture-related queries to seek explanation or clarification.

Given the benefits ChatGPT can provide and the near-infeasibility of massively ban the use of AI-powered software, more regulations should be set up in Chinese academia. Here, more Chinese academic institutions should allow students to use AI-powered software to develop the first draft of their assignments or coursework. However, teachers have to guide students on how to fact-check the details provided by AI and add citations and references accordingly in their coursework. Moreover, teachers should guide students on how to ask AI-powered software questions systematically and creatively, in order to maximize the intellectual outputs generated from ChatGPT. So long as students learn how to capitalize

on the use of ChatGPT, they are able to instrumentalize the software constructively to help enhance their efficiency of academic learning.

## 7. Conclusions and Limitations

Chinese universities lifting the ban on the use of ChatGPT in academic learning will only be a matter of time. More educators are brainstorming and discussing how better state-level, local-level and school-level regulation should be formulated, arranged and delivered in academia. So long as educators find an effective response to evaluate students' academic progress and development, alongside allowing them to use ChatGPT to assist academic learning, AI-powered software will plausibly be further popularized and conducive to students' academic productivity. For example, one of the mainstream approaches adopted by existing higher education institutions is to allow students using ChatGPT but all students' coursework must pass the plagiarism tests. This means while ChatGPT is allowed to be used, students are self-responsible for rephrasing their coursework answers in order to present originality. Moreover, students must be responsible for adding all relevant in-text citations and references in support of the arguments made in their coursework, regardless of whether such points of view are generated by ChatGPT. The inclusion of in-text citations and references whenever arguments are made allows students to maintain their academic rigor.

There are some limitations underlying this research. First, this research project exclusively focuses on analyzing worldwide newspaper articles that investigated the impact of the use of ChatGPT in Chinese academic setting. While such findings should be deemed highly indicative to understand the benefits and risks of using AI-powered device in Chinese academia, newspaper outlets may not necessarily be politically impartial. It is noteworthy that among the 40 studied newspaper articles, six were published by Chinese state-run newspaper outlets (four from *China Daily*; 1 from *CGTN*; 1 from *Global Times*). It is likely that these newspaper articles published by Chinese state-run outlets may be particularly politically partial, resulting in biases and misinformation when presenting the benefits and/or costs of using ChatGPT in Chinese academia. Potentially, if any researchers would like to replicate this research project, they should consider excluding the investigation of any state-run newspaper articles. Alternatively, they should comparatively investigate all eligible newspaper articles and, subsequently, examine only non-state-run newspaper articles. Such a dualistic approach enables researchers to compare whether the exclusion of state-run newspaper articles for examination creates any notable difference in research outputs.

Second, this research project does not apply any systematic review of relevant, available academic research papers. This is because the development of ChatGPT in Chinese academia, to date, is at its preliminary stage. Therefore, there is a significant lack of academic research papers collecting primary data and investigating the benefits and/or risks of using such an AI-powered device in Chinese academia. However, over time, when ChatGPT becomes more developed in Chinese academia and when there is a larger pool of available, relevant research papers evaluating the use of such an AI-powered device on academic learning, we will develop another research project based on applying a systematic review rather than a content analysis. Such an investigation allows the understanding of whether empirical findings developed from the systematic review will be similar to and compatible with those derived from the content analysis.

**Author Contributions:** Conceptualization, J.H.; methodology, J.H.; software, J.H.; validation, J.H., J.C.; formal analysis, J.H.; investigation, J.H.; resources, J.H.; data curation, J.H.; writing—original draft preparation, J.H.; writing—review and editing, J.H.; visualization, J.H.; project administration, J.H.; funding acquisition, J.H. All authors have read and agreed to the published version of the manuscript.

**Funding:** This research received no external funding.

**Institutional Review Board Statement:** Not applicable.

**Informed Consent Statement:** Not applicable.

**Data Availability Statement:** Not applicable.

**Conflicts of Interest:** The authors declare no conflict of interest.

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
