# Peer review of "The Benefits, Risks and Regulation of Using ChatGPT in Chinese Academia: A Content Analysis"

_socsci, doi:10.3390/socsci12070380_

Round 1

Reviewer 1 Report

Notes:

1. The abstract is too long. It is unnecessary to describe the details of the analysis made therein. A brief statement of the purpose of the study, the used methods used and the obtained results is sufficient.

2. It is not indicated on the basis of what criteria you have determined which article falls into the specified category - positive, neutral or negative. It is necessary to clarify how the newspaper articles (the criteria) have been categorized to ensure objectivity.

3. You have researched the number of positive and negative words in the publications - it is advisable to indicate what these words are and provide more in-depth analysis.

4. Is it possible to draw conclusions about the benefits and risks of using ChatGPT in academia by counting the number of positive and negative words in newspaper articles?

Author Response

Dear Reviewer 1,

Thanks for the comments. Please find my responses to your comments, as follows:

  1. The abstract is too long. It is unnecessary to describe the details of the analysis made therein. A brief statement of the purpose of the study, the used methods used and the obtained results is sufficient.

 Response: I trimmed the abstact and now the abstact is under 300 words.

2. It is not indicated on the basis of what criteria you have determined which article falls into the specified category - positive, neutral or negative. It is necessary to clarify how the newspaper articles (the criteria) have been categorized to ensure objectivity.

Response: I added details in lines 200-212 to indicate the basis of the criteria when determining which article falls into the specified category. I also invited an external person to screen all newspaper articles to determine which specified category they fall into. Lines 212-218 provided details on how newspaper articles screening were performed by both the corresponding author and an external screener.

3. You have researched the number of positive and negative words in the publications - it is advisable to indicate what these words are and provide more in-depth analysis.

Response: Lines 359-362 and lines 372-374 indicate what these positive and negative words/phrases are.

4. Is it possible to draw conclusions about the benefits and risks of using ChatGPT in academia by counting the number of positive and negative words in newspaper articles?

Response: The concluding section was expanded. Not only did the corresponding author provide details on how academic regulations on the use of ChatGPT can be realised (lines 444-452), he also added contents about the limitations of his research project (lines 453-479) and how these limitations can be addressed when replicating the research project.

I hope the revised version is on par for publication. Thanks for your review again.

Best,

Reviewer 2 Report

The authors examine benefits and risks of Chinese students who use ChatGPT for academic activities and explores how ChatGPT ought to be regulatored in Chinese academic settings to maintain integrity and ethical standards. According to the research opinions are polarized, with concerns about academic cheating and plagiarism violations, but also recognition of its potential to improve academic quality. 

The paper suggests in which way students can be allowed to use chatgpt and highlights need for regulations and guidelines, and emphasizes the role of educators in teaching students to critically engage with this software.

In my opinion, it provides some insight into the complex landscape of AI-powered tools in academia and offers practical recommendations for their integration.

In Section 4.1. Data Collection, the methodology could be better elaborated, especially visualized as in review papers, emphasizing criteria for selection and analysis process.

Data presented in Table 3 and 4 could be better visualized on a chart. 

The main drowback of the paper is that it does not explicitly address the limitations of the research. 

Especially, potential biases in newspaper articles or the generalizability of findings.

The information about any potential biases in the selection of newspaper articles should be addressed.

Also it should be noted what steps have been taken to mitigate potential biases or misinformation in the articles.

Including other data sources would provide a more diverse range of insights and enrich the findings. Relying solely on newspaper articles may limit the breadh of perspective. 

The conclusion should emphasize and point out the main findings and formulated guidelines. 

Author Response

Dear Reviewer 2,

Thanks for your comments. Please find my responses to your comments, as follows:

In Section 4.1. Data Collection, the methodology could be better elaborated, especially visualized as in review papers, emphasizing criteria for selection and analysis process.

Response: Lines 200-218 added details on how I set up my criteria for selection and analysis process, as well as how I invited an external screener to be involved in independent screening when determining which criteria/category eligible newspaper articles fall into.

Data presented in Table 3 and 4 could be better visualized on a chart. 

Responses: I appreciate the suggestion. Yet, I respectfully disagree the use of charts as charts are often deemed rather unclear graphic outputs for academic presentation.

The main drowback of the paper is that it does not explicitly address the limitations of the research. Especially, potential biases in newspaper articles or the generalizability of findings. The information about any potential biases in the selection of newspaper articles should be addressed.

Responses: Lines 453-468 included details on the potential biases in the selection of newspaper articles and the suggestion of how such biases can be resolved.

Also it should be noted what steps have been taken to mitigate potential biases or misinformation in the articles.

Responses: Both lines 453-468 and lines 469-479 are added paragraphs to help suggest how the problems of potential biases and/or misinformation can be addressed.

Including other data sources would provide a more diverse range of insights and enrich the findings. Relying solely on newspaper articles may limit the breadh of perspective. 

Response: Lines 469-479 suggest that applying a systematic review to examine empirically-researched academic papers should be deemed an alternative to the application of a content analysis to assess newspaper articles. However, the corresponding author indicated that there is a significant lack of research papers (referring to papers that collected primary data rather than simply narrative essays) examining the impacts of ChatGPT on academic learning in Chinese academia. Therefore, only when more empirical research papers are made available, otherwise the corresponding author cannot replicate this research project by applying a systematic review rather than a content analysis.

The conclusion should emphasize and point out the main findings and formulated guidelines. 

Responses: Such details are added on lines 444-452.

Round 2

Reviewer 2 Report

I do not have further suggestions. 

Author Response

Dear Editor,

I have revisited the English language use and believe now it is now in a better form.
